# The Risk of *Clostridioides difficile* Infection in Cirrhotic Patients Receiving Norfloxacin for Secondary Prophylaxis of Spontaneous Bacterial Peritonitis—A Real Life Cohort

**DOI:** 10.3390/medicina57090964

**Published:** 2021-09-13

**Authors:** Irina Girleanu, Anca Trifan, Laura Huiban, Cristina Muzica, Roxana Nemteanu, Andreea Teodorescu, Ana Maria Singeap, Camelia Cojocariu, Stefan Chiriac, Oana Petrea, Sebastian Zenovia, Robert Nastasa, Tudor Cuciureanu, Carol Stanciu

**Affiliations:** 1Gastroenterology Department “Grigore T. Popa”, University of Medicine and Pharmacy, 700111 Iasi, Romania; gilda_iri25@yahoo.com (I.G.); huiban.laura@yahoo.com (L.H.); lungu.christina@yahoo.com (C.M.); maxim_roxxana@yahoo.com (R.N.); andreea.teodorescu19@yahoo.com (A.T.); anamaria.singeap@yahoo.com (A.M.S.); cameliacojocariu@yahoo.com (C.C.); stefannchiriac@yahoo.com (S.C.); stoica_oanacristina@yahoo.com (O.P.); sebastianzenovia20@gmail.com (S.Z.); robertnastasa948@gmail.com (R.N.); drcuciureanutudor@gmail.com (T.C.); stanciucarol@yahoo.com (C.S.); 2Institute of Gastroenterology and Hepatology, “St. Spiridon” University Hospital, 700111 Iasi, Romania

**Keywords:** *Clostridioides difficile* infection, liver cirrhosis, spontaneous bacterial peritonitis, norfloxacin

## Abstract

*Background and Objectives*: Spontaneous bacterial peritonitis (SBP) is a life-threatening complication of liver cirrhosis. Antibiotic prophylaxis is effective but can lead to an increased incidence of *Clostridioides difficile* infection (CDI). The aim of this study was to evaluate the incidence of CDI and the risk factors in cirrhotic patients with a previous episode of SBP receiving norfloxacin as secondary prophylaxis. *Materials and Methods*: We performed a prospective, cohort study including patients with liver cirrhosis and SBP, successfully treated over a 2-year period in a tertiary university hospital. All the patients received secondary prophylaxis for SBP with norfloxacin 400 mg/day. *Results*: There were 122 patients with liver cirrhosis and SBP included (mean age 57.5 ± 10.8 years, 65.5% males). Alcoholic cirrhosis was the major etiology accounting for 63.1% of cases. The mean MELD score was 19.7 ± 6.1. Twenty-three (18.8%) of all patients developed CDI during follow-up, corresponding to an incidence of 24.8 cases per 10,000 person-years. The multivariate Cox regression analysis demonstrated that alcoholic LC etiology (HR 1.40, 95% CI 1.104–2.441, *p* = 0.029) and Child-Pugh C class (HR 2.50, 95% CI 1.257–3.850, *p* = 0.034) were independent risk factors for CDI development during norfloxacin secondary prophylaxis. The development of CDI did not influence the mortality rates in cirrhotic patients with SBP receiving norfloxacin. *Conclusions*: Cirrhotic patients with SBP and Child-Pugh C class and alcoholic liver cirrhosis had a higher risk of developing *Clostridioides difficile* infection during norfloxacin secondary prophylaxis. In patients with alcoholic Child-Pugh C class liver cirrhosis, alternative prophylaxis should be evaluated as SBP secondary prophylaxis.

## 1. Introduction

Spontaneous bacterial peritonitis (SBP) is a severe complication of liver cirrhosis (LC) associated with high mortality [1,2]. SBP occurs in 10–30% of adult cirrhotic patients with ascites and has an in-hospital mortality rate of 20–40% [1,2].

Norfloxacin is a quinolone with limited absorption from the gastrointestinal tract which has antibacterial activity against Gram-negative bacteria. It has been reported that norfloxacin determines gut decontamination and has high efficacy in SBP secondary prophylaxis in patients with LC [2,3].

The EASL [4] and AASLD [5] guidelines recommend that cirrhotic patients with previous episodes of SBP should receive long-term secondary prophylaxis with norfloxacin (400 mg/day) as long as they have ascites. The efficacy of this treatment was first demonstrated by Gines et al. who reported that the use of norfloxacin as secondary prophylaxis for SBP decreased the recurrence of SBP from 68% to 20% [2]. However, there are still controversies regarding the safety of long-term use of norfloxacin. Thus, the European Medicines Agency has released a warning regarding quinolone antibiotics, recommending a restriction in the use of these drugs because of possible side effects such as tendinitis, tendon rupture, hyperglycemia, and aortic aneurysm or dissection [6]. In addition, bacterial resistance should be a major concern in cirrhotic patients, as most of them have had previous hospitalizations and a high probability of being colonized by multidrug resistant bacteria. Moreover, in recent years, an alarm sign has been raised regarding the long-term safety of this treatment regarding the risk of *Clostridioides difficile* infection (CDI). The associated CDI may have a major impact on the outcome of cirrhosis, including higher mortality rates [7].

These data suggest that the use of quinolones for SBP secondary prophylaxis in patients with LC should be reconsidered. The aim of this study was to evaluate the incidence and the risk factors for CDI development in patients receiving long-term norfloxacin for secondary prophylaxis of SBP.

## 2. Materials and Methods

### 2.1. Patients

In this prospective observational cohort study we included all consecutive patients with liver cirrhosis who were hospitalized for SBP between January 2018 and December 2019 for SBP in a tertiary university hospital. The study was approved by the internal review board and Ethical Committee (number 1284/24 November 2017). All the patients were successfully treated for SBP. On the day of hospital discharge, all patients received the recommendation for secondary prophylaxis of SBP. The patients were followed up to a maximum of 24 months, the median follow-up period being 7 months. Follow-up was until the end of the study (December 2019), death or CDI development.

We excluded patients with hepatocellular carcinoma, those who died during hospitalization, patients receiving immunosuppressive drugs, and those with human immunodeficiency virus infection, known hypersensitivity or intolerance to norfloxacin, previous seizure, prior trans-jugular intrahepatic portosystemic shunting, prior solid organ transplantation, prior episodes of SBP, or associated illnesses with a life expectancy of 1 month or who could not be regularly followed-up. All patients were evaluated every 6 months or whenever imposed by LC complications.

Liver cirrhosis diagnosis was based on clinical, laboratory and imaging findings. The severity of liver cirrhosis was evaluated by Child-Pugh class and Model of End-Stage Liver Disease (MELD) score. Ascites was diagnosed by abdominal ultrasound. Comorbidities were recorded for all the patients. All the patients diagnosed with SBP were treated according to the recent guidelines [4,5]. At discharge the patients included in the study received norfloxacin 400 mg/day as secondary prophylaxis of SBP according to the level of their ascites.

### 2.2. Clostridioides Difficile Infection

CDI was diagnosed based on the presence of more than three watery stools within 24 h, plus the presence of *C. difficile* toxins A and/or B (enzyme immunoassay) in stool samples.

Community-associated *CDI* was defined as the onset of CDI outside a healthcare facility or within 48 h following admission to a healthcare facility, without contact from a healthcare facility within the previous 12 weeks [8,9].

Nosocomial CDI was defined as CDI with onset of symptoms on day three or later, following admission to hospital, or in the community within four weeks of discharge from a healthcare facility [9].

Recurrent *CDI* was defined as diarrheal stools with a positive laboratory test more than 2 weeks after end of treatment, and less than 8 weeks following the onset of a previous episode [8,10].

In all patients, an informed written consent to use their clinical data for scientific purposes was systematically obtained at entry. The study was conducted in accordance with the Declaration of Helsinki.

### 2.3. Statistical Analysis

Categorical variables were expressed as frequency and percentage. Continuous variables were expressed as mean plus/minus standard variation for normally distributed continuous data, and as median and range (25th to 75th percentile) to describe non-normally distributed continuous data. Groups were compared using χ2 test for categorical variables, and using independent *t* test or Mann-Whitney *U* test for continuous variables (depending on data distribution). Univariate analysis was performed for each recorded data. Variables with *p* < 0.1 in univariate analysis were included in multivariate analysis (Cox regression). Odds ratio (OR) with 95% confidence interval (CI) was calculated for qualitative variables included in the logistic regression. *p* value < 0.05 was considered statistically significant. All statistical analyses were performed using SPSS 20.0 software (SPSS Inc., Chicago, IL, USA).

## 3. Results

A total of 2520 patients with liver cirrhosis were admitted to our tertiary hospital during the study period, of whom 272 (10.8%) were diagnosed with SBP and only 122 with first episode of SBP were eligible for the study. Most of the patients were males (65.5%), mean age 57.5 ± 10.8 years and had alcoholic liver cirrhosis (63.1%). Baseline characteristics of the patients included in the study are presented in Table 1. The majority of the patients had large ascites (62.3%) and hepatic encephalopathy (85.2%).The mean MELD score was 19.7 ± 6.1 and Child-Pugh class C patients (69.7%) were the most prevalent.

All the patients included in the study were diagnosed with SBP, and the majority (105 patients−86.1%) had community-acquired SBP. All the patients received secondary SBP prophylaxis with norfloxacin 400 mg/day as long as they had ascites. Most of the patients received non-selective beta-blockers (72.1%) and only 26 patients (21.3%) had chronic treatment with PPIs. The median follow-up during the treatment period was 7 months.

Of all the patients included in the study, 23 patients (18.8%) developed CDI during follow-up. The overall CDI incidence rate was 24.8 cases per 10,000 person-years. Three patients were diagnosed with healthcare-associated CDI. These were diagnosed with CDI during hospitalization for hepatic encephalopathy. Six patients (26.1%) had recurrent form of CDI. All the patients received treatment with vancomycin 125 mg every 6 h orally for 10 days. In ten patients the dose of vancomycin was increased to 250 mg every 6 h as they did not response to the initial dose. Four patients with recurrent CDI received the tapering vancomycin regimen. Fidaxomicin was used in one patient that relapsed after tapering with vancomicin.

There was no significant difference in the presence of hepatic encephalopathy or hepato-renal syndrome, the concomitant use of beta-blockers (BB) or proton pump inhibitors (PPIs), as well as for most of the laboratory parameters, between cirrhotics that developed CDI and those without (Table 1). The majority of the patients (85.2%) received rifaximin for hepatic encephalopathy prophylaxis. However, the patients that developed CDI during follow-up were predominantly males (86.9% vs. 60.6%, *p* = 0.017), had significantly higher Child-Pugh score (12 points vs. 10 points, *p* = 0.016) and more frequent alcoholic etiology of LC (78.3% vs. 59.6%; *p* = 0.030). No previous reported quinolone side effects were identified in our cohort.

Eighteen patients (14.8%) had SBP recurrence during follow-up. The development of CDI infection did not influence SBP recurrence rate. During follow-up 66 patients (54.1%) died. The mortality rate was similar in patients with or without CDI (60.8% vs. 52.5%, *p* = 0.496). During the study period the median number of admissions was also similar between the two groups (1.91 ± 1.41 vs. 2.36 ± 1.50; *p* = 0.192). Twelve patients (9.8%) received antibiotic treatment during follow-up, with no significant difference between the study groups (8.7% vs. 10.1%; *p* = 0.838). Seven patients received Ciprofloxacin for urinary tract infection, three patients received Amoxicillin for respiratory tract infection, and two patients received cefotaxime for acute cholecystitis and lobar pneumonia. None of the patients received liver transplant during the follow-up.

The results of the univariate and multivariate logistic regression analyses are shown in Table 2. The males with alcoholic LC, Child-Pugh C class and large ascites were at risk of developing CDI during follow-up. The multivariate Cox regression analysis demonstrated that alcoholic LC etiology (HR 3.18, 95% CI 1.104–2.441, *p* = 0.029) and Child-Pugh C class (HR 2.50, 95% CI 1.257–3.850, *p* = 0.034) were independent risk factors for CDI development during norfloxacin secondary prophylaxis for SBP.

## 4. Discussion

Previous studies have evaluated the incidence of CDI in patients with liver cirrhosis and concluded that it is higher than in the general population [7]. Patients with LC are prone to develop infection complications especially in the advanced form of this disease. Previous hospitalization, immuno-compromised system, comorbidities and most importantly previous antibiotic treatment were the main risk factors for CDI development in patients with LC. It was also demonstrated that the development of CDI in cirrhotic patients is associated with an increased risk of mortality, prolonged hospitalization and higher hospitalization costs [7].

SBP is a severe complication of LC associated with a high risk of mortality. The secondary prophylaxis with norfloxacin has proven to be effective in these patients [2]. Gut decontamination by eliminating the aerobic gram-negative bacilli reduces the rate of SBP recurrence caused by *Enterobacteriaceae* [11]. Norfloxacin is a quinolone with low permeability and solubility, characteristics associated with selectivity in bowel decontamination. Moreover, in vitro studies demonstrated that norfloxacin could have an anti-inflammatory effect by decreasing the level of TNF-alfa, explaining the positive effect on mortality [12,13,14].

Even if the efficacy of this treatment was clearly demonstrated, questions regarding its safety were raised recently. Several Food and Drug Administration Drug Safety Communications [15] preceded the European Medicines Agency alert [6]. The fluoroquinolone/quinolone use was associated with significant hypoglycemia and side effects involving the tendons, muscles, joints, and nerves with an increased risk of developing tendinitis and tendon rupture. Moreover, an increased incidence of aortic aneurysm or dissection secondary to quinolone use was demonstrated [16,17]. Considering all these side effects concerns have been raised regarding the safety of long-term administration of norfloxacin in patients with LC. The aim of our study was to evaluate the incidence and the risk factors associated with the CDI development in patients with LC receiving norfloxacin as secondary prophylaxis of SBP.

In our cohort, the incidence of CDI cirrhotic patients was 24.8 cases per 10,000 person-year, lower than the incidence reported in a cohort of cirrhotic patients with HE [18]. Previous studies demonstrated that in the general population treated with antibiotics, 5% of the patients developed CDI [19]; in our cohort 6.7% developed CDI during norfloxacin secondary SBP prophylaxis. It should be also mentioned that the majority of the studies on CDI epidemiology in LC were retrospective and use different methods of CDI diagnosis, with a high risk of underestimating the real incidence of this disease [20]. Most of the patients developed CDI infection after 6 months of norfloxacin prophylaxis and the presence of CDI did not influence mortality. The patients with alcoholic liver cirrhosis and Child-Pugh class C had a significant higher risk of developing CDI during follow-up. Sundaram et al. also reported a significantly higher prevalence of CDI among patients with alcoholic liver disease compared to those without alcoholic liver disease (1.62% vs. 1.04%, *p* < 0.001) [21] and these data were confirmed by our study. The recurrence rate was only 14.8%, comparable with the data previously reported [2]. It has to be mentioned that most of the patients included in our study also received rifaximin for hepatic encephalopathy prophylaxis. Even if Bajaj et al. demonstrated that cirrhotic patients with CDI have a higher mortality rate and length of hospital stay compared with those without CDI [7], in our cohort the CDI infection did not influence the mortality rate. These data are comparable with data obtained in patients with LC, but without SBP [22].

Most of the randomized controlled trials (RCT) did not report CDI as complication of long-term antibiotic prophylaxis for variceal hemorrhage [3] or SBP [11,23,24]; moreover, norfloxacin was associated with reduced *Clostridium spp* in patients’ stools [2], although, when compared to other antibiotic classes, but not to placebo, norfloxacin was associated with CDI in 9.7% of cases compared to no cases in the co-trimoxazole group [25].

Our real-world data partially confirmed the data from RTC. In the RCT, Norfloxacin was found to reduce the probability of recurrence of SBP, as Gines et al. demonstrated [2,26], and these data were confirmed by our study, although the risk of CDI still remained in our cohort.

Our study has some strengths and also several limitations. Thus, it is the first study that had as primary outcomes the evaluation of the incidence and risk factors for developing CDI in cirrhotic patients receiving long-term norfloxacin prophylaxis for SBP. However, as a single center study it is more likely to produce bias secondary to the small number of cases or underdiagnosed CDI in some cases. In addition, the study provides no information on the *C. difficile* strain.

## 5. Conclusions

Patients with Child-Pugh class C alcoholic liver cirrhosis have a high risk of developing CDI during long-term norfloxacin treatment for SBP secondary prophylaxis. For these patients, alternative prophylaxis should be evaluated. SBP secondary prophylaxis should be personalized in order to outweigh the risks associated with the CDI development.

## Figures and Tables

**Table 1 medicina-57-00964-t001:** The characteristics of the study groups.

Parameter	All Patients *n* = 122	Patients with CDI *n* = 23	Patients without CDI *N* = 99	*p*
Age, years, Mean ± SD	57.5 ± 10.8	57.8 ± 11.8	57.4 ± 10.7	0.874
Mean follow-up, months, Median, (IQR)	7 (2–15)	7 (2.5–15.5)	7 (2–14.5)	0.638
Gender, male, *n* (%)	80 (65.5)	20 (86.9)	60 (60.6)	**0.017**
Etiology of cirrhosis, *n* (%)				**0.030**
Alcohol	77 (63.1)	18 (78.3)	59 (59.6)	
HBV	26 (21.3)	5 (21.7)	21 (21.2)	
HCV	19 (15.6)	0 (0)	19 (19.2)	
Child-Pugh class, *n* (%)				**0.045**
B	37 (30.3)	3 (13.1)	34 (34.3)	
C	85 (69.7)	20 (86.9)	65 (65.7)	
Child-Pugh score, Median, (IQR)	10 (9–12)	12 (10–12.5)	10 (9–12)	**0.016**
MELD, Mean ± SD	19.7 ± 6.1	21.6 ± 7.9	19.3 ± 5.5	0.095
Grade 3 ascites, *n* (%)	76 (62.3)	18 (78.2)	58 (58.6)	0.079
ALT, UI/L, Median, (IQR)	33 (24–65)	44 (31–66)	32 (24–63)	0.290
Platelets × 10^5^, Median, (IQR)	118 (78–142)	100 (76–128)	122 (79–144)	**0.046**
INR, Median, (IQR)	1.62 (1.4–1.95)	1.9 (1.47–2.36)	1.6 (1.38–1.9)	0.059
Total bilirubine, mg/dl, Median, (IQR)	4.2 (1.79–7.72)	7.28 (2.18–8.44)	4.06 (1.79–7.08)	0.203
Albumin g/dl, Median, (IQR)	2.19 (1.93–2.77)	2.26 (1.98–2.9)	2.17 (1.9–2.7)	0.282
Creatinine mg/dl, Median, (IQR)	0.79 (0.64–1.22)	0.81 (0.76–1.53)	0.77 (0.63–1.18)	0.211
CRP, g/dl, Median, (IQR)	2.9 (1.8–6.7)	2.7 (2.1–7.9)	2.9 (1.7–6.1)	0.736
SBP recurrence, *n* (%)	18 (14.8)	6 (26.1)	12 (12.1)	0.089
Death, *n* (%)	66 (54.1)	14 (60.8)	52 (52.5)	0.496
PPIs, *n* (%)	26 (21.3)	5 (21.7)	21 (21.2)	0.956
BBs, *n* (%)	88 (72.1)	18 (78.2)	70 (70.7)	0.467
Rifaximin, *n* (%)	104 (85.2)	20 (86.9)	84 (84.8)	0.797
Number of admissions during follow-up, Mean ± SD	2.27 ± 1.48	1.91 ± 1.41	2.36 ± 1.50	0.192
Antibiotic treatment during follow-up, *n* (%)	12 (9.8)	2 (8.7)	10 (10.1)	0.838
Total proteins ascites, Median, (IQR)	1.1 (0.8–1.6)	1.1 (0.8–1.46)	1.1 (0.8–1.65)	0.760

CDI: *Clostridioides difficile* infection; SD: standard deviation; IQR: interquartile range; HBV: hepatitis B virus; HCV: hepatitis C virus; MELD: Model for End-Stage Liver Disease; ALT: alanine aminotransferase; INR: International Normalized Ratio; CRP: C-reactive protein, SBP: spontaneous bacterial peritonitis; PPIs: proton pump inhibitors; BB: beta-blockers. The bold values are statistically significant.

**Table 2 medicina-57-00964-t002:** Risk factors for CDI—univariate and multivariate analyses.

Parameter	Univariate Analysis			Multivariate Analysis		
	OR	95%CI	*p*-Value	HR	95%CI	*p*-Value
Alcoholic etiology	1.38	1.082–1.775	0.030	3.18 2.50	1.104–2.441 1.257–3.850	**0.029** **0.034**
Child-Pugh C class	1.32	1.070–1.639	0.033			
Age > 65 years	1.23	0.558–2.715	0.615			
Males	1.43	1.147–1.795	**0.011**			
Grade 3 ascites	1.33	1.018–1.753	**0.045**			
Comorbidities	0.56	0.273–1.153	0.069			
Ascitic liquid proteins < 1.5 g/L	1.01	0.724–1.406	0.959			
Rifaximin	1.02	0.857–1.226	0.795			
PPIs	1.10	0.862–1.421	0.458			

CDI: *Clostridioides difficile* infection; OR: odds ratio; CI: confidence interval; HR: hazard ratio; PPIs: proton pump inhibitors. The bold values are statistically significant.

## Data Availability

Data supporting reported results can be provided as request in an electronic format.

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
