# Peer review of "The Risk of Clostridioides difficile Infection in Cirrhotic Patients Receiving Norfloxacin for Secondary Prophylaxis of Spontaneous Bacterial Peritonitis—A Real Life Cohort"

_medicina, 2021, doi:10.3390/medicina57090964_

Round 1

Reviewer 1 Report

The present study reports the incidence in a prospective cohort of patients receiving norfloxacin as secondary prophylaxis for SBP. Findings are not new and it has several aspects that should be clarified. My comments:

  • It would be interesting to have a control group not receiving norfloxacin. It would help at clarifying the potential role of norfloxacin for CDI. Many previous studies do not point at that.
  • Important follow-up events that may contribute to CDI development are not accounted for. CDI is associated with hospital admissions (for HE, bleeding or whatever) and with previous AB treatments, specially for broad spectrum AB. Authors must report number of admissions and use of antibiotics other than norflo and rifaximin. I think that including these variables could add to explain the so-high prevalence of CDI in this cohort. It is mandatory to include in the analysis intermediate admissions as well as the use of AB during follow-up others than norflo/rifaximine.
  • Plenty of patients receiving rifaximin, it looks like overprescribed... Was it strictly indicated for recurrent HE? If so, it could translate many previous hospital admissions and therefore a non-considered risk factor for CDI. Please clarify.
  • TIPS patients excluded, why? Despite PHT can be decreased many of these patients persist with ascites and therefore at risk for SBP. I would not exclude them.
  • Statistical analysis: I would only plan a temporal analysis since not all events are occurring at the same time point. It is not correct to define analyze “death during follow-up” in Table 1 since not all patients are followed-up for the same time-period, did they? If so please specify the follow-up period. Anyway, I would suggest to use a temporal analysis (avoid logistic regression unless pre-specified timepoints for each patient).
  • Compenting events analysis: I suppose that many patients died or underwent transplantation, so they could not develop CDI infection. For this case the statistical approach should be a competing risks analysis where transplantation or death are competing with CDI infection, a patients who dies or undergoes transplantation cannot develop CDI. Also, as previously commented, intermediate admissions and AB treatments must be included in the analysis since probably related with CDI incidence.
  • Did any patient undergo transplantation? Please clarify.
  • Survival plot: it is not clear if follow-up is from inclusion or from CDI infection. It should be from inclusion to the study. The effect of CDI on survival should be analyzed as a time-dependent or modifying factor during follow-up.
  • What was the ascites protein concentration? Did it influence outcome and CDI incidence?
  • Was fidaxomicin used for any relapse?
  • Why authors propose in their conclusions alternatives to norfloxacin if survival is not affected by CDI? It is difficult to save an effective and safe AB for long term treatments... So, reasons to change/substitute them should be robust and strong with a good alternative. Maybe norflo is the less bad AB for this purpose...

Author Response

We thank the Reviewer for considering our manuscript. Below are Reviewer 1’s specific comments and our response to each one.

The present study reports the incidence in a prospective cohort of patients receiving norfloxacin as secondary prophylaxis for SBP. Findings are not new and it has several aspects that should be clarified. My comments:

  1. It would be interesting to have a control group not receiving norfloxacin.It would help at clarifying the potential role of norfloxacin for CDI. Many previous studies do not point at that.

Response: Thank you for this suggestion. Considering the fact that SBP secondary prophylaxis is recommended by the international guidelines and our national and local protocol for SBP treatment it will be unethical not to give norfloxacin secondary prophylaxis in patients with liver cirrhosis and SBP.

  1. Important follow-up events that may contribute to CDI development are not accounted for. CDI is associated with hospital admissions (for HE, bleeding or whatever) and with previous AB treatments, specially for broad spectrum AB. Authors must report number of admissions and use of antibiotics other than norflo and rifaximin. I think that including these variables could add to explain the so-high prevalence of CDI in this cohort. It is mandatory to include in the analysis intermediate admissions as well as the use of AB during follow-up others than norflo/rifaximine.

Response: We thank the Reviewer for these remarks. We reviewed the number of admissions during the follow-up and there were no significant differences between the two study groups (please see page 6 line 26 and table 1). Also, during follow-up 12 patients received antibiotic treatment in both groups, with no significant difference between the study groups (please see page 4 paragraph 1 and table 1).

  1. Plenty of patients receiving rifaximin, it looks like overprescribed...Was it strictly indicated for recurrent HE? If so, it could translate many previous hospital admissions and therefore a non-considered risk factor for CDI. Please clarify.

Response: We fully agree with the Reviewer ‘s comments. The patients included in our study were in the decompensated stage of liver cirrhosis, already diagnosed with HE. The rifaximin was used for HE prevention as the current guidelines are recommended.

  1. TIPS patients excluded, why?Despite PHT can be decreased many of these patients persist with ascites and therefore at risk for SBP. I would not exclude them.

Response- TIPS is decreasing PHT and the risk of SBP and also decrease mortality and morbidity, the number of hospital admissions, and all these could represent confounding factors when mortality is one of the end-points. In our screened cohort no patient had previous TIPS.

  1. Statistical analysis: I would only plan a temporal analysis since not all events are occurring at the same time point.It is not correct to define analyze “death during follow-up” in Table 1 since not all patients are followed-up for the same time-period, did they? If so please specify the follow-up period. Anyway, I would suggest to use a temporal analysis (avoid logistic regression unless pre-specified time points for each patient).

Response- We fully agree with the Reviewer‘s comments. The patients were follow-up maxim 24 months, median follow-up period was 7 months. The follow-up was until the end of the study (December 2019), death or CDI development (please see page 2). Figure 1 was removed. The Table 1 was changed (please see table 1).

  1. Competing events analysis:I suppose that many patients died or underwent transplantation, so they could not develop CDI infection. For this case the statistical approach should be a competing risks analysis where transplantation or death are competing with CDI infection, a patients who dies or undergoes transplantation cannot develop CDI. Also, as previously commented, intermediate admissions and AB treatments must be included in the analysis since probably related with CDI incidence.

ResponseWe agree with these comments but perhaps did not make this clearer in the original text. No patient received a liver transplant during follow-up. In prospective cohort studies death is always a competing event when analyze the incidence of an event (CDI development). There were no differences between the two groups regarding intermediate admissions or AB treatment during the follow-up (please see page 4, paragraph 1 and Table 1).

  1. Did any patient undergo transplantation?Please clarify.

Response: None of the patients received liver transplantation during the follow-up (please see page 4, paragraph 1).

  1. Survival plot: it is not clear if follow-up is from inclusion or from CDI infection. It should be from inclusion to the study. The effect of CDI on survival should be analyzed as a time-dependent or modifying factor during follow-up.

Response- the follow-up started from the inclusion in the study (when the patients started the norfloxacin secondary prophylaxis). Considering your valuable suggestions we removed from the manuscript the survival plot. 

  1. What was the ascites protein concentration?Did it influence outcome and CDI incidence?

Response- the protein concentration in the ascites did not influenced the CDI incidence. The data were added in table 1.

  1. Was fidaxomicin used for any relapse?

Response- Fidaxomicin was used in one patient that relapsed after tapering with Vancomicin (please see page 3).

  1. Why authors propose in their conclusions alternatives to norfloxacin if survival is not affected by CDI?It is difficult to save an effective and safe AB for long term treatments... So, reasons to change/substitute them should be robust and strong with a good alternative. Maybe norflo is the less bad AB for this purpose...

Response Norfloxacin is an effective and safe antibiotic in patients with liver cirrhosis. In our study norfloxacin did not influenced the mortality rate although increased the morbidity rate. This is the reason that we propose alternatives to norfoxacin.

Reviewer 2 Report

In the submitted manuscript titles “The risk of Clostridioides difficile infection in cirrhotic patients receiving Norfloxacin for secondary prophylaxis of spontaneous bacterial peritonitis- a real life cohort”, Girleanu et al. report that SBP and Child-Pugh C class and alcoholic liver can be risk factors for the development of CDI during long-term norfloxacin treatment in cirrhotic patients. They also recommend looking for alternatives for the prophylactic application in patients with alcoholic Child-Pugh C class liver cirrhosis. The study is interesting. I just ask the authors to go through the manuscript to check for typos. Also mention the ethical approval at the beginning of materials and methods section.

Author Response

We thank the reviewer for the comments on our manuscript. These is our respons point by point:

In the submitted manuscript titles “The risk of Clostridioides difficile infection in cirrhotic patients receiving Norfloxacin for secondary prophylaxis of spontaneous bacterial peritonitis- a real life cohort”, Girleanu et al. report that SBP and Child-Pugh C class and alcoholic liver can be risk factors for the development of CDI during long-term norfloxacin treatment in cirrhotic patients. They also recommend looking for alternatives for the prophylactic application in patients with alcoholic Child-Pugh C class liver cirrhosis. The study is interesting.

 I just ask the authors to go through the manuscript to check for typos.

Response : This has been done (please see the revised manuscript).

Also mention the ethical approval at the beginning of materials and methods section.

Response : This has been done (please see page 2).

Round 2

Reviewer 1 Report

Authors present an almost identic version of the original manuscript. Major changes, including a temporal regression analysis have been obviated. So, my recommendation is the same as in the previous version.